# Tea Consumption Patterns in Relation to Diet Quality among Children and Adults in the United States: Analyses of NHANES 2011–2016 Data

**DOI:** 10.3390/nu11112635

**Published:** 2019-11-03

**Authors:** Florent Vieux, Matthieu Maillot, Colin D. Rehm, Adam Drewnowski

**Affiliations:** 1MS-Nutrition, 27 bld Jean Moulin Faculté de Médecine la Timone, Laboratoire C2VN, 13385 Marseille CEDEX 5, France; florent.vieux@ms-nutrition.com (F.V.); Matthieu.maillot@ms-nutrition.com (M.M.); 2Department of Epidemiology & Population Health Albert Einstein College of Medicine, Montefiore Medical Center, Bronx, NY 10595, USA; colin.rehm@gmail.com; 3Center for Public Health Nutrition, University of Washington, Box 353410, Seattle, WA 98195, USA

**Keywords:** tea, beverages, consumption, demographics, socioeconomic status, diet quality, Nutrient-Rich Foods (NRF9.3), Healthy Eating Index 2015, CVD biomarkers, BMI

## Abstract

Flavonoid-rich tea offers an alternative to sugar-sweetened beverages. The present analyses, based on 2 24-hour dietary recalls for 17,506 persons aged ≥9 years old in the 2011–2016 National Health and Nutrition Examination Survey database (NHANES 2011–2016), explored tea consumption patterns in relation to demographics, diet quality, cardiovascular disease (CVD) biomarkers (lipids and blood pressure), and body weight. Beverage categories were unsweetened tea, other tea (herbal and presweetened tea), coffee, milk, 100% juice, water and other high-calorie (HC) and low-calorie (LC) beverages. Tea consumption (18.5% of the sample) was highest among older adults (51–70 years old), non-Hispanic Asians and Whites, and those with college education and higher incomes. The effects of age, gender, education, income, and race/ethnicity were all significant (*p* < 0.001 for all). Adult tea consumers had diets with more protein, fiber, potassium, iron, and magnesium, and less added sugars and alcohol. Their diets contained fewer HC beverages and coffee but had more total and citrus fruit, more total dark green and orange vegetables, and more seafood, eggs, soy and milk. Tea consumers had higher Healthy Eating Index (HEI-2015) and higher Nutrient-Rich Foods (NRF9.3) nutrient density scores. Few children drank tea and no differences in diet quality between consumers and non-consumers were observed. Adult tea consumers had slightly higher high-density lipoprotein (HDL) cholesterol and lower body mass index (BMI) values. Tea consumption was associated with higher socioeconomic status and better diets.

## 1. Introduction

Other than water, tea is the most frequently consumed beverage worldwide [1,2]. The most commonly consumed teas are black tea (fermented), oolong (semi-fermented), and green tea (unfermented) [1,2]. Daily tea consumption patterns can vary greatly by geography, from a reported high of 788 g/day (~4.3 cups) in the UK to a low of 14 g/day (~0.1 cups) in Navarra, Spain [3]. Analyses of global consumption patterns have suggested that coffee consumption follows per capita country-level incomes, whereas tea consumption does not [1].

Relatively little is known about tea consumption patterns across socio-demographic groups in the US [4]. In past analyses of National Health and Nutrition Examination Survey (NHANES) data, beverage consumption followed a socio-economic gradient [5,6,7]. Water and diet beverages were associated with higher education and incomes; the opposite trend was observed for sugar-sweetened beverages and fruit-based drinks [5,6]. In the 1999–2002 NHANES, tea consumers were more likely to be female, older, non-Hispanic White, and had higher incomes than non-consumers [8]. That early study also noted that tea was the major source of flavonoids in the US diet [8].

Flavonoids found in tea, chocolate, red wine, fruit, and vegetables have long been associated with multiple health benefits [9,10,11]. Tea, wine, beer, citrus fruits, and apples are the most important dietary sources of flavonoids among US adults [12]. The principal flavonoids in tea are flavanols (catechin, epicatechin) and flavonols [8,13]. Some observational studies have noted an inverse association between higher tea consumption and lower blood pressure [9,14] and reduced cardiovascular disease (CVD) risk [15], likely due to higher flavonoid intakes [10,15]. Green tea catechins were associated with lower blood pressure and lower low-density lipoprotein (LDL) cholesterol [16] and with healthier aging [17]. Flavonoid-rich foods, including tea, contribute to lower total and cause-specific mortality [13,18]. 

The present study explored tea consumption patterns in relation to diet quality and selected health outcomes across different sociodemographic groups in the US. Analyses were based on multiple cycles (2011–2016) of NHANES. The NHANES is the flagship US program for monitoring population diet and is the common evidence base for dietary recommendations and guidelines.

## 2. Materials and Methods 

### 2.1. Dietary Intake Databases

Consumption data for tea and other beverages, including water, came from 3 cycles of the nationally representative National Health and Nutrition Examination Survey (NHANES), corresponding to years 2011–2012, 2013–2014, and 2015–2016 [19]. The three NHANES cycles provided a total sample of 29,902 persons—of whom, 23,013 were aged ≥ 9 years. Out of these, 5507 individuals were removed because they completed 1 day recall only. The present analyses were based on a total of 17,506 persons who were aged ≥9 years and had completed 2 days food recalls. 

The NHANES 24 hour recall uses a multi-pass method, where respondents reported the types and amounts of all food and beverages consumed in the preceding 24 hours, from midnight to midnight. The multi-pass method was conducted by a trained interviewer using a computerized interface [20]. Respondents first identified a quick list of foods and beverages consumed. The time and consumption occasion for each food item was also obtained. A more detailed cycle then recorded the amounts consumed, followed by a final probe for any often-forgotten foods (beverages, condiments). Survey participants 12 years and older completed the dietary interview on their own. Proxy-assisted interviews were conducted with children 6–11 years of age, where a parent or guardian with knowledge of the child’s diet was present and able to assist. 

Day 1 interviews were conducted by trained dietary interviewers in a mobile examination center. Day 2 interviews were conducted by telephone some days later [21]. The mean number of days between dietary recalls was 7.9 days (median 5 days).

### 2.2. Participant Characteristics

NHANES participants were stratified by gender and age. The age group cut-points were: 9–13 years, 14–19 years, 20–30 years, 31–50 years, 51–70 years, and > 70 years. These age groups generally correspond to the age groups used by the Institute of Medicine to examine Dietary Reference Intakes. Race/ethnicity was defined as non-Hispanic white; non-Hispanic black, Mexican American, Other Hispanic, non-Hispanic Asian, and other/mixed race. Family income-to-poverty ratio (IPR) is the ratio of family income to the federal poverty threshold; the cut-points for IPR were <1; 1–1.99; 2–3.49; and ≥3.5. Obesity was defined as BMI > 30 based on measured heights/weights (body mass index = kg/m^2^).

### 2.3. Consumption of Tea and Beverages

The NHANES 24 hours recall data provide the amounts in grams of each food and beverage consumed [19]. Water and beverages were classified into 8 categories: tea (excluding herbal and pre-sweetened teas); other tea including herbal teas or infusions and pre-sweetened teas, milk and milk beverages, coffee, 100% juice, water, other high-calorie (HC) beverages with > 50 kcal/8 oz., and other low-calorie (LC) beverages with < 50 kcal/8 fl. oz. Drinking water included both bottled and tap water.

### 2.4. Diet Quality Indicators

The USDA Food and Nutrient Database for Dietary Studies (FNDDS) was used to calculate the energy content of the diet, based on caloric beverages and solid foods [22]. This information was supplemented with data from the Food Patterns Equivalents Database (FPED) from the United States Department of Agriculture (USDA) [23] in order to estimate the intakes of meaningful food groups (e.g., vegetables or fruit) and to calculcate summary measures of diet quality. 

The Healthy Eating Index (HEI-2015) is the latest iteration of the USDA diet quality measurement tool, specifically designed to monitor compliance with the 2015 Dietary Guidelines for Americans [24]. The HEI-2015 is an energy adjusted summary measure of diet quality based on the intake of 9 food groups/nutrients to encourage including total fruits, whole fruits, total vegetables, greens and beans, whole grains, dairy, total protein foods, seafood and plant protein, and fatty acids ratio, and 4 food groups/nutrients to discourage, including refined grains, sodium, added sugars and saturated fat. 

The Nutrient-Rich Foods (NRF9.3) index, applied to total diets, was the second measure of diet quality [25]. The NRF9.3 is based on two subscores: NR and LIM. NR is composed on 9 nutrients to encourage whereas LIM is based on 3 nutrients to limit. Reference daily values (DVs) were based on the US Food and Drug Administration (FDA) and other standards. The 9 NR nutrients to encourage and standard reference amounts were as follows: protein (50g), fiber (28g), vitamin A (900 RAE), vitamin C (90 mg), vitamin D (20 mcg), calcium (1300 mg), iron (18 mg), potassium (4,700 mg) and magnesium (420 mg). The 3 LIM nutrients to limit and maximum recommended values (MRVs) were: added sugar (50g), saturated fat (20g), and sodium (2,300 mg). The NRF9.3 was calculated as follows:**NRF9.3 = (NR − LIM) × 100**
with
(1)NR=∑i=19IntakeiEnergy×2000DVi
(2)LIM=∑i=13IntakeiEnergy×2000MRVi−1

Intake_*i*_ is the daily intake of each nutrient i and DVi is the reference daily value for that nutrient. In NR calculation. each daily nutrient intake i was adjusted for 2000 kcal and expressed in percentage of DV. Following past protocol, percent DVs for nutrients were truncated at 100, so that an excessively high intake of one nutrient could not compensate for the dietary inadequacy of another. In calculating LIM, only the share in excess of the recommended amount was considered. 

### 2.5. Data Availability and Ethical Approval

The necessary Institutional Review Board (IRB) approval for NHANES had been obtained by the National Center for Health Statistics (NCHS) [26]. Adult participants provided written informed consent. Parental/guardian written informed consent was obtained for children. Children/adolescents ≥ 12 years provided additional written consent. All NHANES data are publicly available on the NCHS and USDA websites [27]. All documentation of laboratory methodology, including plasma lipid analyses, is provided online at wwwn.cdc.gov. Per University of Washington (UW) policies, public data do not involve “human subjects” and require neither IRB review nor an exempt determination. Such data may be used without any involvement of the Human Subjects Division or the UW Institutional Review Board.

### 2.6. Statistical Analyses

The survey-weighted mean intakes of tea and all other beverages were evaluated overall and by age group, gender, race/ethnicity, family income-to-poverty ratio and education. Tea consumption was assessed by estimating the percentage of tea consumers and the amounts of tea consumed by age group, gender, race/ethnicity, family income-to-poverty ratio and education. The prevalence of obesity (body mass index >30) was also calculated for each subpopulation reported. Average amounts (in grams/day) and contributions to total beverage intakes (in %) of tea, other tea (including herbal infusions), coffee, milk, 100% juice, water and HC and LC beverages were assessed by age, race/ethnicity and education. Tea consumers were classified by tertiles of tea consumption (T1, T2, T3). Four tea consumption categories were defined: non consumers, small consumers (T1), medium consumers (T2) and high consumers (T3). Nutrient intakes of tea consumers and non-consumers were adjusted per 2000 kcal/day dietary energy and were compared using general linear models adjusting for gender, age, ethnicity, IPR (and education in adults). HEI 2105 scores for tea consumers and non-consumers were also compared using general linear models adjusting for gender, age, ethnicity, IPR (and education in adults). Similar analyses were conducted comparing consumption of selected food groups by tea consumers versus non-consumers. Finally, adult tea consumers and non-consumers were compared on a variety of CVD biomarkers, including plasma lipids, blood pressure and BMI values. 

Differences between proportions were tested using chi-square tests and differences in continuous variables were tested using generalized linear models. All analyses accounted for the complex survey design of NHANES and reflect dietary behaviors of the US adult population from 2011–2016. All analyses were conducted using SAS software version 9.4 (SAS Institute Inc., Cary, NC, USA) and SURVEYREG, SURVEYMEANS and SURVEYFREQ procedures.

## 3. Results

### 3.1. Tea Consumption by Age and Socio-Demographics

Tea consumers were defined as those NHANES participants who were drinking tea only on day one (5.36% of sample), only on day two (6.17% of sample), or on both days (6.96% of sample). The percentage of tea consumers in the total sample was 18.49%. Among adults aged >19 years the percentage consuming tea was 20.77%.

Table 1 shows the patterns of tea consumption by socio-demographics. Percentage of obese NHANES respondents by socio-demographics is presented as well. First, females were more likely to be tea consumers than were males; the effect of gender was significant (*p* < 0.0001). Second, the percentage of tea consumers more than doubled with age (*p* < 0.0001). One in four adults aged >50 years in the NHANES sample consumed tea but fewer than 1 in 10 children or adolescents. Third, tea drinking varied among racial/ethnic groups. Least likely to drink tea were Mexican Americans and non-Hispanic Blacks. Most likely to drink tea were non-Hispanic Asians and non-Hispanic Whites.

The percent of tea consumers approximately doubled with rising education and incomes. The effects of income-to-poverty ratio (IPR) and educational attainment were both significant (*p* < 0.0001). Analyses of amounts of tea consumed produced a similar socio-demographic profile of tea consumers. There were significant effects of age, race/ethnicity, education and incomes. Most tea was consumed by adults >50 years, non-Hispanic Asians and Whites, and by groups with IPR > 3.5.and college education. 

Obesity prevalence data, also shown in Table 1**,** showed the expected effects of gender, age, race/ethnicity, education, and incomes. In univariate analyses, higher obesity prevalence was observed among women, older adults (>50 years), non-Hispanic Blacks, and among groups of lower education and incomes.

### 3.2. Tea and Other Beverage Consumption by Socio-Demographics

Figure 1 shows how tea and other beverage consumption patterns changed with age. Figure 1 Top shows amounts in g/day, while Figure 1 Bottom shows the percentage contribution of each beverage category to the total. The consumption of tea and herbal tea increased with age, as did the consumption of coffee. Milk and juice consumption declined with age. The consumption of HC beverages was highest among adolescents and young adults but declined sharply with age. By contrast the consumption of LC beverages increased with age.

Figure 2 shows tea and all other beverage consumption patterns by race/ethnicity. Figure 2 Top shows amounts in g/day, while Figure 2 Bottom shows the percentage contribution of each beverage category to the total. Non-Hispanic Asians consumed most tea, most drinking water, and the lowest amounts of HC beverages. Non-Hispanic Whites consumed most coffee and most LC beverages. Non-Hispanic Blacks and Mexican Americans consumed least tea and most HC beverages.

Figure 3 shows tea and all other beverage consumption patterns for adults only by education. Figure 3 Top shows amounts in g/day, while Figure 3 Bottom shows the percentage contribution of each beverage category to the total. Groups with lowest education consumed least drinking water, least tea and most HC beverages. Groups with highest education consumed most water, most tea, and least HC beverages. Those groups also consumed more coffee and LC beverages.

### 3.3. Tea Consumption in Relation to Other Beverages

Table 2 shows other beverage consumption patterns among adults by tertiles of tea consumption expressed in cups per day. The tertile cut-points were <0.85; 0.85 to 1.77; and >1.77 cups/day. In adjusted models, higher amounts of tea consumed were associated with lower amounts of coffee and less milk, but also significantly lower amounts of HC beverages, mostly sugar-sweetened beverages. There were no significant differences between tea consumers and non-consumers in the consumption of water, other tea (including herbal infusions), juice or LC beverages.

### 3.4. Tea Consumption and Diet Quality

Table 3 shows energy and nutrient intakes for tea consumers versus non-consumers. Data are presented separately for children (9–19 years) and for adults (>19 years). Among children, there were no significant differences in energy or nutrient intakes between tea consumers and non-consumers. Among adults, there was no difference in daily energy intakes; however, diets of adult tea consumers were significantly higher in protein, dietary fiber, vitamin E, potassium, iron and magnesium. Diets of adult tea consumers had significantly higher NR and NRF9.3 nutrient density scores compared to non-consumers. 

Whereas the NRF9.3 is a nutrient density score, HEI-2015 is a measure of compliance with food-based Dietary Guidelines for Americans 2015–2020. Table 4 shows differences in HEI-2015 scores for the whole NHANES sample and separately for children and adults. HEI-2015 measures were based on two 24 h recalls and analyses were adjusted for covariates. Tea drinking among adults (but not among children) was associated with higher HEI-2015 scores, adjusting for dietary energy, age, gender, ethnicity, education and incomes.

Table 5 shows HEI-2015 sub-scores of interest for NHANES adults. Compared to non-consumers, adult tea consumers consumed more total fruit, citrus fruit and other fruit; more vegetables, dark green vegetables, and orange and other vegetables, and more seafood, eggs, soy and milk, and oils. Adult tea consumers also consumed less added sugars and less alcohol. However, their dietary patterns also contained more starchy vegetables and refined grains.

### 3.5. Tea Consumption in Relation to Health Outcomes

Table 6 shows associations between tea consumption and selected biomarkers. Blood draws for lipid analyses were based on fasting participants tested in the morning. For those analyses, the sample size was accordingly reduced. As shown, tea consumption was associated with higher high-density lipoprotein (HDL) cholesterol. There were no significant differences between tea consumers and non-consumers in terms of plasma triglycerides, LDL cholesterol, or blood pressure. There was a very weak but significant negative association between tea consumers and body weight: tea consumers had lower BMI values.

Correlation analyses examined the relation between amount of tea consumed (among consumers only) and cholesterol, triglyceride, and blood pressure outcomes. A weakly significant (*p* = 0.048) negative relation between tea consumption and direct HDL cholesterol was observed.

## 4. Discussion

As both research and dietary guidance shift from individual nutrients to composite food patterns [28,29], the relation between tea drinking and diet quality metrics is waiting to be explored. Tea can have a direct effect through bioactive compounds, but it can also be an indicator of, if not a vector for, healthier diets. 

For the total NHANES sample aged ≥9 years, 18.5% were tea consumers. The prevalence of tea consumers was higher among adults >19 years (20.77%) and increased sharply with age: approximately 26% of adults aged >51 years drank tea. Song and Chum [8] noted in 2003 that 21.3% of US adults aged >19 years reported drinking tea daily; those numbers were comparable to those observed here. In parallel with the 1999–2002 HNANES [8], tea drinking was associated with higher education and incomes and was most common among non-Hispanic Asians and non-Hispanic Whites. Based on comparisons with other studies on beverage consumption patterns [5], that sociodemographic profile was counter to that observed with HC beverages. The consumption of sugar-sweetened beverages in the US generally tracks lower education and incomes.

Tea consumption among adults in the 2011–2016 NHANES was associated with higher diet quality scores, measured using NRF9.3 and HEI-2015 indices. The NRF9.3 index is nutrient-based and was adjusted per 2000 kcal to assess dietary nutrient density [25]. Tea consumers had more nutrient-rich diets with more protein, vitamins and minerals, and less added sugars. The HEI-2015 was designed to assess compliance with the 2015–2020 Dietary Guidelines and is mostly food based. Adult tea consumers’ higher HEI-2015 scores were largely due to a higher consumption of fruit, vegetables, seafood, and oils and lower consumption of added sugars and alcohol. 

As shown in Table 3, adult tea consumers had significantly lower intakes of HC beverages and had diets with significantly less added sugar (Table 5). Coffee consumption was reduced among the top two tertiles of tea consumers by volume and milk among the top tertile only. The consumption of plain water, juices, diet beverages, or other tea (including herbal infusions) was not different across the two groups. Arguably, tea consumption was a characteristic of, if not a vector for, healthier beverage consumption patterns.

The flavonoid content of tea deserves a mention. The USDA nutrient composition databases now include bioactives and antioxidants [30,31], one major dietary source of which is tea. The relative healthfulness of different beverages has been explored before [32,33]. For example, the Beverage Guidance Panel [32] used the previously developed nutrient profiling approach to rank beverages according to their energy and nutrient contents. Water was ranked highest in order of preference and was followed by tea and coffee, low-fat (1.5% or 1%) and skim (nonfat) milk and soy beverages, and then low-calorie (LC) beverages, fruit and vegetable juices, whole milk, alcohol, and sports drinks, and finally by high-calorie (HC) sugar-sweetened beverages [32]. Even though tea and coffee were ranked second behind water, they were still viewed as problematic. The authors cautioned that the addition of milk, cream, or caloric sweeteners to tea or coffee would increase their energy density and lower nutritional value [32]. A version of beverage guidelines for Mexico [33] assigned beverages into 6 categories from most to least healthy: The levels were water, skim or low-fat milk, coffee and tea without sugar; LC beverages, HC beverages, and sugar-sweetened beverages [33]. In other words, the nutrient density of beverages was not based on their content of vitamins, minerals or bioactives but on the absence of calories, total or added sugars or total or saturated fat. 

Measures of beverage nutrient density that rely on the absence of sugar and fat are themselves problematic. Popkin et al. [32] did suggest that the potential health benefits of flavonoids in tea ought to be more fully explored. Since then, studies have examined the impact of tea consumption on health outcomes [34,35,36]. Greyling [9] found a small effect in a meta-analysis of 11 pooled studies. Habitual tea consumption was associated with better health-related quality of life in older adults in China [37]. Another study reported a small reduction in blood pressure (SBP 2.36; DBP 1.77) [38]. Black tea also lowered LDL cholesterol especially among subjects at risk for CVD [36].

Recent activities by the US Food and Drug Administration (FDA) provide a further illustration of the shift in dietary guidance to from nutrients to foods, food groups and dietary ingredients [28,29]. The FDA’s current definition of a healthy food is based on nutrient content per serving. The criteria include those for nutrients to limit (fat, sugar, and salt) and those for nutrients to encourage, including vitamin A, vitamin C, calcium, iron, protein, and fiber. The criteria are linked to elements in the Nutrition Facts label and serving size regulations. The 2015 citizen petition by KIND Inc, requested that the FDA revisit the definition of what constitutes a “healthy” food. The KIND bars, which contained nuts, did not meet the nutrient content claim for “healthy” because they contained more than 1 g of saturated fat per Reference Amount Customarily Consumed and because >15% of energy came from saturated fat. The KIND petition argued that nutrient density was more important than low fat content. Although the FDA has long favored the nutrient-based approach, certain food groups or ingredients may well be recognized as intrinsically healthy by the agency. Tea drinking may be one characteristic of healthy food patterns, along with whole grains, whole fruit and nuts and seeds. In that case, there is a need to adapt current nutrient profiling methods to facilitate federal regulations, recommendations and guidelines. 

The present study had limitations. The sample size for <19 years old was limited by low tea consumption among children and adolescents, the groups most likely to consume SSBs. Because of the cross-sectional study design, any associations between tea consumption and obesity, BMI or health outcomes (blood pressure and lipid levels) should be interpreted cautiously and no causal inferences should be made. While NHANES dietary intakes were based on self-report, the scale and representativeness of the NHANES sample make it the premier study of dietary intakes in the US and the foundation of food and nutrition policy.

## 5. Conclusions

Adult tea consumers had more nutrient-rich diets, containing more desirable food groups and more nutrients to encourage. Tea drinking, associated with higher socioeconomic status, was also associated with better dietary choices. In particular, tea consumption was associated with a significantly lower consumption of HC beverages and of added sugars.

## Figures and Tables

**Figure 1 nutrients-11-02635-f001:**
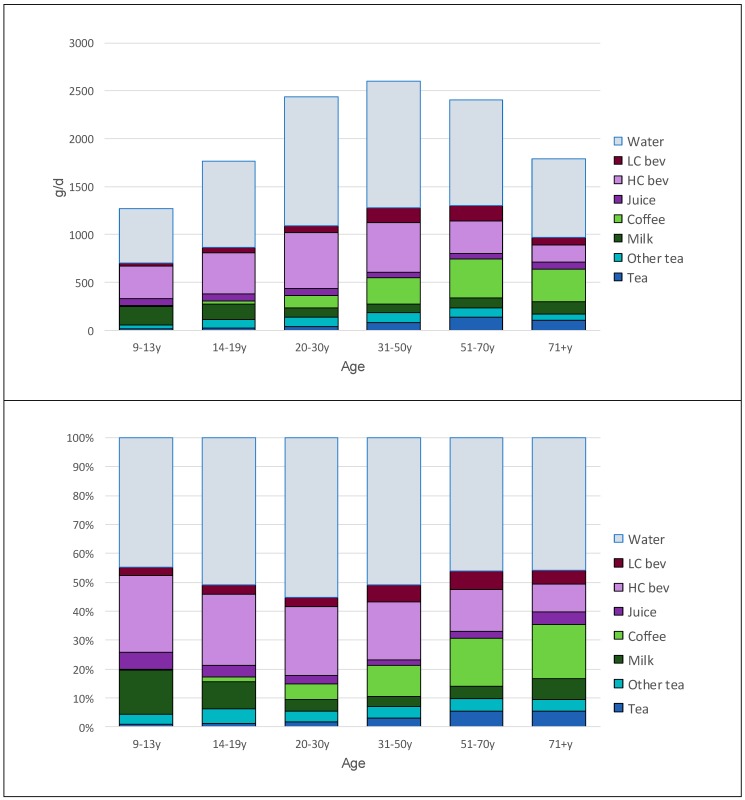
Tea and beverage consumption patterns by age group. Top panel shows amounts in g/day, while bottom panel shows percentages of total beverage consumption by beverage category.

**Figure 2 nutrients-11-02635-f002:**
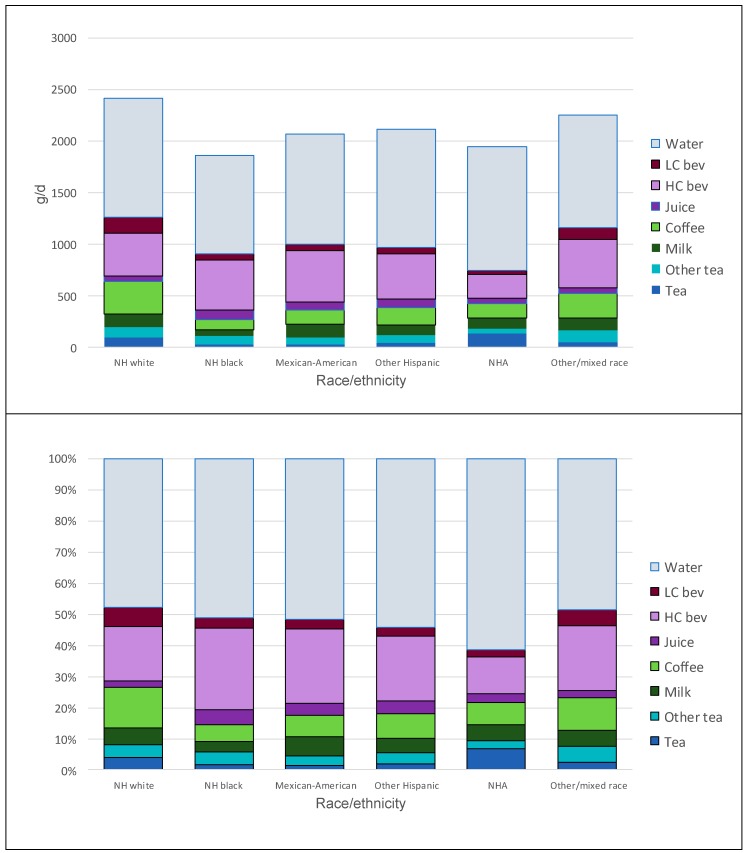
Tea and beverage consumption patterns by race/ethnicity. Top panel shows amounts in g/day, while bottom panel shows percentages of total beverage consumption by beverage category.

**Figure 3 nutrients-11-02635-f003:**
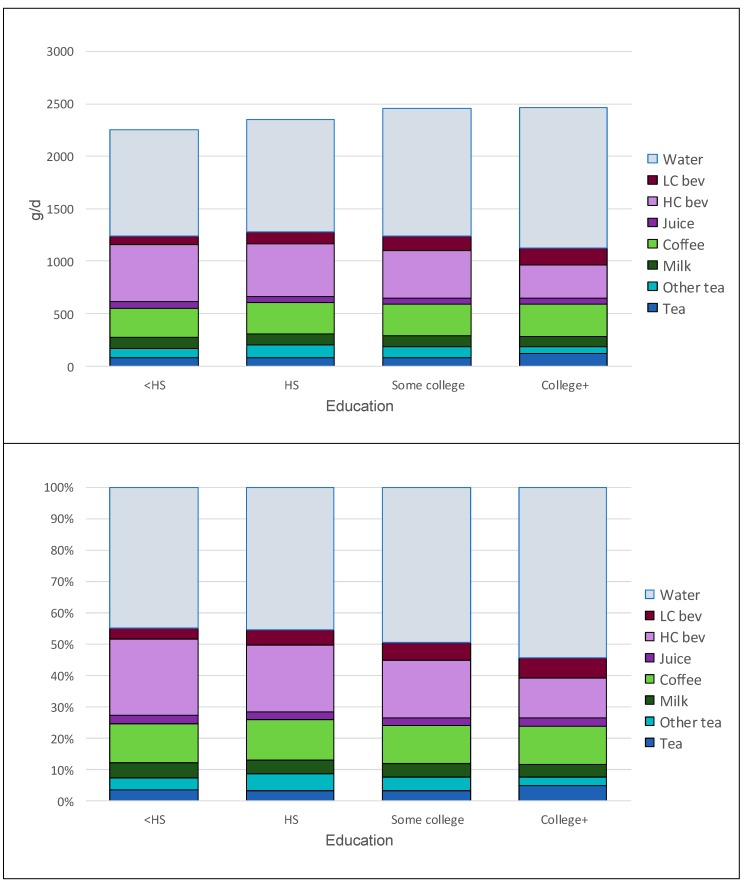
Tea and beverage consumption patterns among adults by education. Top panel shows amounts in g/day, while bottom panel shows percentages of total beverage consumption by beverage category.

**Table 1 nutrients-11-02635-t001:** Tea consumers (%), amounts consumed (grams and SD), and obesity prevalence (% and CI) by socio-demographics.

		N	Consumers %	Amounts Grams (SD)	Obesity Prevalence % [Confidence Intervals]
Gender	Male	8425	16.06	75.13 (6.72)	31.43 [29.68;33.18]
	Female	9081	20.73	82.34 (5.36)	35.33 [33.53;37.13]
	Test		<0.0001	0.2266	<0.0001
Age (year)	9–13	2176	5.21	10.11 (2.04)	5.76 [4.20,7.31]
	14–19	2339	8.04	20.85 (3.81)	17.84 [14.89,20.78]
	20–30	2370	11.91	39.67 (5.51)	30.57 [27.77,33.36]
	31–50	4391	17.73	74.83 (5.51)	39.32 [36.95;41.70]
	51–70	4368	26.67	133.55 (12.42)	41.26 [38.28;44.25]
	71+	1862	26.98	98.87 (6.91)	32.77 [29.81;35.73]
	Test		<0.0001	<0.0001	<0.0001
Race/Ethnicity	Non-Hispanic White	6351	20.44	95.82 (7.15)	32.58 [30.82;34.34]
	Non-Hispanic Black	4120	10.88	29.44 (3.21)	42.33 [40.04;44.62]
	Mexican American	2656	10.32	29.73 (3.41)	38.02 [35.53;40.51]
	Other Hispanic	1827	12.98	40.66 (5.73)	35.15 [32.54;37.75]
	Non-Hispanic Asian	1872	34.20	131.41 (9.83)	11.55 [9.73;13.37]
	Other-Mix	680	16.56	52.93 (13.24)	36.81 [30.43;43.18]
			<0.0001	<0.0001	<0.0001
Family income-to-poverty ratio (IPR)	<1	4008	10.55	42.6 (4.37)	34.55 [32.32;36.77]
	1–1.99	4262	16.05	63.16 (7.53)	36.89 [34.24;39.55]
	2–3.49	3327	18.49	73.48 (7.12)	33.30 [31.08;35.52]
	3.5+	4554	23.31	107.43 (9.18)	31.04 [28.39;33.69]
	Missing	1355	17.62	68.24 (10.25)	34.68 [30.68;38.67]
	Test		<0.0001	<0.0001	0.002
Education (adults only)	<High school	2679	15.66	76.05 (16.05)	40.88 [38.68;43.08]
	High school	2852	17.62	78.29 (8.14)	41.13 [37.55;44.72]
	Some college	3994	19.18	80.7 (7.82)	42.22 [39.51;44.93]
	College+	3459	26.75	116.87 (8.63)	28.77 [26.15;31.40]
	Test		<0.0001	0.0015	<0.0001

**Table 2 nutrients-11-02635-t002:** Beverage consumption among adults (g/day) by tea consumption status. Non-consumers (NC) are compared to consumers split by tertiles (T1, T2, and T3) of tea consumption (Least square means).

	Tea Non-Consumers (N = 10,386)	Tea Consumers (N = 2605)			
		T1 [0;0.85]	T2 [0.85;1.77]	T3 > 1.77	*p*-Value
Tea	−2.87	130.00	285.5	874.05	<0.0001 ^a,b,c^
Other tea	85.31	86.83	87.25	95.21	0.9083
Milk	95.37	99.83	82.70	70.99	0.0018 ^c^
Coffee	261.30	226.01	200.24	111.97	<0.0001 ^b,c^
Juice	69.82	63.93	57.02	63.68	0.0801
Water	1131.92	1157.15	1054.27	1115.73	0.2041
HC beverages	443.30	328.79	348.07	309.94	<0.0001 ^a,b,c^
LC beverages	89.60	79.85	65.50	65.73	0.4682

Values adjusted for energy intakes, age, sex, race/ethnicity, education and IPR. Significant 2 by 2 difference between NC and T1 marked ^a^; Significant 2 by 2 difference between NC and T2 marked as ^b^; Significant 2 by 2 difference between NC and T3 marked as ^c^.

**Table 3 nutrients-11-02635-t003:** Energy and nutrient intakes (adjusted for 2000 kcal) and Nutrient-Rich Foods (NRF9.3) scores and NR9 and LIM subscores for tea consumers and non-consumers shown for children and adults.

	Children/Teenagers				Adults			
	Non Consumers (N = 4229)	Consumers (N = 286)	*p*-Value (Unadjusted)	*p*-Value Adjusted *	Non Consumers (N = 10,386)	Consumers (N = 2605)	*p*-Value (Unadjusted)	*p*-Value Adjusted *
Energy (kcal/d)	1999	1914	0.0917	0.0397	2092	1977	<0.0001	0.2633
Protein (g/2000 kcal)	76.06	78.36	0.2678	0.3919	80.80	83.39	0.0015	0.0066
Total sugar (g/2000 kcal)	117.36	113.94	0.1782	0.2696	105.53	98.28	<0.0001	0.0003
Added sugar (tsp/2000 kcal)	17.83	17.59	0.7575	0.7800	15.80	12.94	<0.0001	<0.0001
Saturated fat (g/2000 kcal)	25.49	25.56	0.9023	0.9108	24.50	24.18	0.1849	0.2234
Sodium (mg/2000 kcal)	3410	3537	0.0870	0.2113	3425	3511	0.0039	0.0403
Fiber (g/2000 kcal)	15.26	16.58	0.0083	0.0098	17.16	19.60	<0.0001	<0.0001
Vitamin D (mcg/2000 kcal)	5	5	0.2032	0.1951	5	5	0.0169	0.4788
Vitamin A (mcg/2000 kcal)	618	755	0.5244	0.4779	666	726	0.0042	0.9421
Vitamin E (mg/2000 kcal)	7	8	0.6477	0.8068	9	10	<0.0001	0.0009
Vitamin C (mg/2000 kcal)	74	75	0.8483	0.5167	85	94	0.0004	0.1731
Calcium (mg/2000 kcal)	1060	1053	0.8716	0.7620	961	960	0.9300	0.3830
Potassium (mg/2000 kcal)	2326	2419	0.0784	0.0946	2669	2963	<0.0001	<0.0001
Iron (mg/2000 kcal)	15	16	0.6927	0.6249	14	15	<0.0001	0.0008
Magnesium (mg/2000 kcal)	253	271	0.0085	0.0177	303	333	<0.0001	<0.0001
NR9 subscore	550.89	556.25	0.5744	0.4428	559.24	597.20	<0.0001	<0.00001
LIM subscore	79.55	85.04	0.1146	0.2814	77.54	80.14	0.0666	0.2627
NRF9.3 score	471.34	471.21	0.9898	0.7522	481.70	517.05	<0.0001	0.0004

All measures are the average of 2 days recalls. * adjusted for gender, age, ethnicity, IPR (and education in adults).

**Table 4 nutrients-11-02635-t004:** Healthy Eating Index (HEI-2015) diet quality scores for tea consumers and non- consumers for children and adults. Univariate and adjusted models are presented.

**All**	**Non-consumers (N = 14,615)**	**Consumers (N = 2891)**	***p*-value**
HEI-2015	50.51	54.89	<0.0001
Adjusted for gender, age, ethnicity, and IPR *	50.96	53.18	<0.0001
Adjusted for gender, age, ethnicity, IPR, and energy *	50.96	53.18	<0.0001
**Children**	**Non-consumers (N = 4229)**	**Consumers (N = 286)**	***p*-value**
HEI-2015	46.74	47.25	0.6472
Adjusted for gender, age, ethnicity, and IPR *	47.70	48.25	0.6079
Adjusted for gender, age, ethnicity, IPR, and energy *	47.70	48.21	0.6313
**Adults**	**Non-consumers (N = 10,386)**	**Consumers (N = 2605)**	***p*-value**
HEI-2015	51.37	55.37	<0.0001
Adjusted for gender, age, ethnicity, IPR, and education *	51.94	53.90	<0.0001
Adjusted for gender, age, ethnicity, IPR, education, and energy *	51.94	53.90	<0.0001

* Least square means.

**Table 5 nutrients-11-02635-t005:** Amounts of food groups consumed (units provided) among adult tea consumers and non-consumers.

	Unadjusted			Adjusted Model 1			** Adjusted Model 2		
	Non Consumers (N = 10,386)	Consumers (N = 2605)	*p*-Value	Non Consumers (N = 10,386)	Consumers (N = 2605)	*p*-Value *	Non Consumers (N = 10,386)	Consumers (N = 2605)	*p*-Value **
Total fruit (cups/day)	0.96 (0.02)	1.13 (0.04)	0.0003	0.95 (0.02)	1.14 (0.04)	<0.0001	1.04 (0.04)	1.15 (0.04)	0.0233
Citrus (cups/day)	0.21 (0.01)	0.30 (0.02)	0.0004	0.21 (0.01)	0.30 (0.02)	0.0002	0.24 (0.03)	0.29 (0.03)	0.0292
Other (cups/day)	0.48 (0.02)	0.61 (0.02)	<0.0001	0.48 (0.01)	0.62 (0.02)	<0.0001	0.51 (0.02)	0.59 (0.02)	0.0073
Juice (cups/day)	0.26 (0.01)	0.22 (0.02)	0.0091	0.26 (0.01)	0.22 (0.02)	0.0356	0.30 (0.01)	0.27 (0.02	0.163
Total vegetables (cups/day)	1.52 (0.02)	1.73 (0.03)	<0.0001	1.50 (0.02)	1.75 (0.03)	<0.0001	1.47 (0.02)	1.62 (0.04)	0.0002
Dark greens (cups/day)	0.16 (0.01)	0.21 (0.01)	0.0003	0.16 (0.01)	0.21 (0.01)	0.0003	0.16 (0.01)	0.19 (0.01)	0.0469
Red orange (cups/day)	0.39 (0.01)	0.43 (0.01)	0.0053	0.38 (0.01)	0.43 (0.01)	<0.0001	0.36 (0.01)	0.39 (0.01)	0.0205
Other (cups/day)	0.55 (0.01)	0.66 (0.02)	<0.0001	0.54 (0.01)	0.67 (0.02)	<0.0001	0.54 (0.01)	0.61 (0.02)	0.002
Total grains (oz/day)	6.44 (0.05)	6.34 (0.09)	0.321	6.24 (0.04)	6.49 (0.06)	0.0007	6.54 (0.05)	6.84 (0.07)	0.0001
Whole grains (oz/day)	0.96 (0.02)	1.11 (0.03)	0.0007	0.94 (0.02)	1.12 (0.03)	<0.0001	0.93 (0.03)	0.10 (0.04)	0.1184
Refined grains (oz/day)	5.48 (0.05)	5.23 (0.09)	0.0077	5.30 (0.04)	5.36 (0.06)	0.3515	5.60 (0.05)	5.84 (0.06)	0.0005
Protein foods (oz/day)	6.19 (0.06)	6.14 (0.12)	0.7173	6.03 (0.05)	6.26 (0.11)	0.043	6.26 (0.07)	6.47 (0.11)	0.063
Eggs (oz/day)	0.55 (0.01)	0.58 (0.02)	0.2336	0.54 (0.01)	0.59 (0.02)	0.0339	0.57 (0.01)	0.63 (0.02)	0.0315
Soy (oz/day)	0.08 (0.01)	0.11 (0.01)	0.007	0.08 (0.01)	0.11 (0.01)	0.0036	0.07 (0.01)	0.09 (0.01)	0.0138
Nuts/seeds (oz/day)	0.78 (0.04)	0.91 (0.04)	0.0017	0.75 (0.03)	0.94 (0.04)	<0.0001	0.68 (0.04)	0.72 (0.04)	0.3866
Total legumes (oz/day)	0.12 (0.00)	0.11 (0.01)	0.1048	0.12 (0.00)	0.12 (0.01)	0.5213	0.16 (0.01)	0.16 (0.01)	0.952
Total dairy (cups/day)	1.58 (0.02)	1.45 (0.03)	<0.0001	1.53 (0.02)	1.48 (0.03)	0.0863	1.36 (0.02)	1.30 (0.03)	0.0398
Oils (g/day)	26.14 (0.30)	26.89 (0.40)	0.108	25.30 (0.23)	27.51 (0.34)	<0.0001	24.81 (0.28)	26.28 (0.42)	0.0022
Solid fats (g/day)	34.88 (0.37)	31.99 (0.60)	<0.0001	33.61 (0.23)	32.92 (0.48)	0.1509	32.00 (0.27)	31.49 (0.41)	0.24
Added sugars (tsp/day)	16.86 (0.25)	13.21 (0.32)	<0.0001	16.27 (0.23)	13.65 (0.29)	<0.0001	15.76 (0.27)	14.19 (0.30)	<0.0001
Alcohol (servings/day)	0.71 (0.02)	0.47 (0.03)	<0.0001	0.67 (0.02)	0.50 (0.03)	<0.0001	0.53 (0.02)	0.35 (0.03)	<0.0001

* Least square means (adjusted for energy); ** Least square means (adjusted for gender, ethnicity, IPR, age, education, and energy).

**Table 6 nutrients-11-02635-t006:** Cardiovascular disease biomarkers and body weight among adults among tea consumers and non-consumers.

	N	Non-Consumers	Consumers	*p*-Value (Unadjusted)
High-density lipoprotein (HDL) Cholesterol				
Direct HDL Cholesterol (mg/dL)	12,416	53.49 (0.34)	55.41 (0.51)	0.0013
Direct HDL Cholesterol (mmol/L)	12,416	1.38 (0.01)	1.43 (0.01)	0.0012
Triglyceride and LDL cholesterol				
Triglyceride (mg/dL)	5900	121.7 (1.86)	125.88 (3.71)	0.3097
Triglyceride (mmol/L)	5900	1.37 (0.02)	1.42 (0.04)	0.3096
LDL cholesterol (mg/dL)	5817	113.99 (0.87)	115.75 (1.22)	0.2235
LDL cholesterol (mmol/L)	5817	2.95 (0.02)	2.99 (0.03)	0.2237
Blood pressure				
Average systolic blood pressure (mmHg)	12,725	122.21 (0.33)	123.52 (0.66)	0.0617
Average diastolic blood pressure (mmHg)	12,691	70.54 (0.25)	70.41 (0.4)	0.7097
Body weight				
Body mass index	12,711	29.29 (0.13)	28.70 (0.28)	0.0389

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
