# Peer review of "Tea Consumption Patterns in Relation to Diet Quality among Children and Adults in the United States: Analyses of NHANES 2011–2016 Data"

_nutrients, 2019, doi:10.3390/nu11112635_

Round 1
Reviewer 1 Report
The manuscript entitled “Tea consumption patterns in relation to diet quality among children and adults in the United States: Analyses of NHANES 2011-16 data”
is a research paper on a current topic which will draw attention to readers from many audiences.
Please after the authors consider the following comments:
Major comments
Please comment on whether there is information from the NHANES data on whether the tea consumed was black, green or oolong tea, hot or iced. Do we have any information on whether sugar or honey or/and milk or cream were added to tea consumed? If not, please comment in relation to findings reported in present manuscript (eg Table 3) or in other studies (eg Lines 276-279). You may wish to use throughout the manuscript (text, tables and figures) the term “herbal infusions” instead of the terms “other tea” (eg Table 2) or “herbal teas” (eg Line 172) or “infusions” (eg Figures 1,2,3). Please comment in the discussion on the potential effect of seasonality on the outcome of the analysis. Do we have any information on whether data were collected on summer or winter? Please briefly explain in the methodology section how data on BMI and CVD biomarkers were obtained. Lines 267-268. Please further explain (or rephrase in current manuscript) the argument that tea may be a vector for healthier diets.
Minor comments
Please revise the title in Table 3, in order to mention that NRF 9.3 scores and subscores are presented. Please revise the title in Table 5, in order to better explain content and to reflect that HEI-2015 subscores were used (and not any other method of dietary pattern evaluation). Please revise y axis title in Figures 1B,2B, and 3B (g/d must be a typo) Line 293. There is a typo (to from).
Reviewer 2 Report
This is a nicely written paper, that explores a topic that has rarely been studied; however, the authors should give emphasis on the need of papers like this one in the Introduction and the Discussion sections, mainly by improving their Results. In other words, authors should answer to the Q "Why should i read this paper?"
Author Response
Thank you for a positive review.
Reviewer 3 Report
This paper describes tea consumption using the NHANES 2011-2016 data. The methods, particularly statistical methods, are not adequately described making it very difficult to assess the merits of the manuscript. Manuscript should also be revised to read more like a paper for a scientific journal. In it's current form it is too informal.
Abstract
The abstract is not structured properly.
Line 14: should be a comma, not a period after tea.
Line 17: Did not define first “NH”.
Income does not need to be plural.
Line 41: It is not clear what “social gradients” refers to.
Line 47-48: Consider revising this sentence to “Some observational studies have observed an inverse association between higher tea consumption and lower blood pressure…”. Additionally, references are needed at the end for “reduced CVD risk”.
Line 55: Missing a period at the end of the sentence.
Line 61: There should be a comma, not period for 29,902.
Line 68-69: Not entirely accurate, participants 12+years answered for themselves. Interviews of children ages 9-11, answered themselves, but with the assistance of a proxy familiar with the child’s intake.
Line 71: Provide the average amount of days between recalls, “some days later” is not useful.
Line 72: should be a comma, not period for 17,506.
Line 80: What is the justification for they family income to poverty ratio cut-off selected? Additionally, researchers need to add how they defined obesity. All cut-offs for covariates should be in the methods section.
Line 123: NR was not defined and there is a random period after the word “calculation”.
Line 126: LIM was not defined and there is a random period after “LIM”.
Line 131: What NHANES data is available on the USDA website? FNDDS is not NHANES data.
Methods (in general): Authors should add information on response rates, any exclusions that were made, missing data, etc. Need more information on how final analytic sample was determined.
Line 145-148: How is it possible that the % of tea consumers on each recall day was 5% and 6%, respectively, but the percentage of tea consumers in the total sample was 18%? What is meant by the total sample?
Line 152: It is not appropriate to state “increased several-fold”. State clearly the actual results.
Results (in general): Section needs to be re-written to sound more like a scientific research paper.
Table 1: What is in the parenthesis under amounts column? Standard error?
Table 1: Authors need to provide a measure of variability with prevalence estimates – e.g., standard error, 95% confidence interval.
Figures 1a and 1b/2a and 2b/3a and 3b: Details on how this analysis was done needs to be in the methods section.
Table 2: This table is completely confusing to try and follow.
Table 3/4/5: The analyses needs to be described in greater detail in the methods, statistical analysis section.
Table 5: Some measure of variability needs to be included.
Table 6: Researchers must describe the variables included in table 6 in the methods section – how were they collected, what were the laboratory methods used? Additionally, are these continuous variables all normally distributed? Was normality assessed? Authors never mentioned correlation analyses in the methods.
Round 2
Reviewer 2 Report
This is a very interesting and well written paper in which authors try to explore tea consumption patterns in relation to diet quality and selected health outcomes across different socio-demographic groups in the United States. However, it needs again some revisions:
Line 15: CVD should be defined. Line 22: Put “,” instead of “.” after fruit. Line 79-80: The authors state that the two principal flavonoids in tea are flavonols and flavonols. Please check. Line 103: The sentences in lines 103-105 need to be rephrased. Line 159: NR and LIM should be defined. Line 176: The index “i” should be united with intake. Line 182: IRB should be defined. In Subsection 2.6 the statistical significance level should be mentioned. The 2ndparagraph of Subsection 3.1 should be rephrased. The language is incomprehensible and the expression “the percentage of tea drinkers more than doubled with age” is not justified by Table 1. On Table 1, NH, NHA and HS should be defined. Line 330: “Sex” should be replaced with “gender”. On Table 2, the star symbol should be defined. Line 488: SSB should be defined.Author Response
Please see the attachment

Reviewer 3 Report
This manuscript has been significantly improved. There are still some original comments that were not adequately addressed by the authors and some new ones.
Reviewer’s original statement: Line 80: What is the justification for they family income to poverty ratio cut-off selected? Additionally, researchers need to add how they defined obesity. All cut-offs for covariates should be in the methods section.
Response: The goal for IPR cutoffs was to divide the sample into four approximately equal groups for maximum power of analysis. In other studies with a smaller sample we used 3 IPR groups and the cutoffs were <1.3; 1-3-3.49, and >3.5. Obesity is now defined in text terms of BMI.
Reviewer’s follow-up to authors’ response: That’s fine, but authors should state that they split income into quartiles based on the data.
Reviewer’s original statement: Methods (in general): Authors should add information on response rates, any exclusions that were made, missing data, etc. Need more information on how final analytic sample was determined.
Response: The methodology of the National Health and Nutrition Survey, conducted by the federal government of the United States has been very extensively documented over the years on government websites and other publications. The complete federal database is provided to researchers free of charge. We have no insight into response rates, exclusions, missing data etc.
Reviewer’s follow-up to authors’ response: NHANES publishes response rates on their website for each NHANES cycle (https://wwwn.cdc.gov/nchs/nhanes/ResponseRates.aspx). Response rates are very important for the reader – usually this sort of information is included in manuscripts. Many published papers using NHANES data include response rates and many journals actually require that information to be included.
I don’t know why the authors have no insight or knowledge on the missing data from the data they are analyzing. Using a simple example with the dietary data, there is a variable called “Dietary recall status” that provides information on why data may be missing. Reviewer is just asking for more information on how the final analytical sample was arrived at. The new information provided in lines 95-86 is sufficient.
Reviewer’s original statement: Line 152: It is not appropriate to state “increased several-fold”. State clearly the actual results.
Response: that would depend on age group. We can say “more than doubled”.
Reviewer’s follow-up to authors’ response: You could also try adding in some estimates. For example: There was a linear increase in tea consumption with increasing age, from 5.2% in children aged 9-13 y to about 27% in older adults aged 71+ y (p<0.0001).
Reviewer’s original statement: Results (in general): Section needs to be re-written to sound more like a scientific research paper.
Response: The results are described very simply. Here is a sample of the writing – it does sound like scientific paper.
Obesity prevalence data, also shown in Table 1, showed the expected effects of gender, age, race/ethnicity, education, and incomes. In univariate analyses, higher obesity prevalence was observed among women, older adults (>50y), non-Hispanic Blacks, and among groups of lower education and incomes.
Reviewer’s follow-up to authors’ response: This paper has improved significantly since last version. It’s nice to see things like confidence intervals and standard errors in the tables. The results do not provide estimates from the results in the tables or figures. This is a decision that needs to be made by the editor of the journal. If the journal prefers to not have any estimates in the results section of the paper – that is fine.
Reviewer’s original statement: Table 1: Authors need to provide a measure of variability with prevalence estimates – e.g., standard error, 95% confidence interval.
Response: Confidence intervals are provided
Reviewer’s follow-up to authors’ response: Is there a reason why a standard error or confidence interval can’t be provided for the prevalence of tea consumers?
Reviewer’s original statement: Table 6: Researchers must describe the variables included in table 6 in the methods section – how were they collected, what were the laboratory methods used? Additionally, are these continuous variables all normally distributed? Was normality assessed? Authors never mentioned correlation analyses in the methods.
Response: The documentation for the NHANES procedures is provided online at:
https://wwwn.cdc.gov/Nchs/Nhanes/2013-2014/TRIGLY_H.htm
Laboratory methods are provided on the website as well. For example:
Description of Laboratory Methodology
This method is based on the work by Wahlefeld using a lipoprotein lipase from microorganisms for the rapid and complete hydrolysis of triglycerides to glycerol followed by oxidation to dihydroxyacetone phosphate and hydrogen peroxide. The hydrogen peroxide produced then reacts with 4-aminophenazone and 4-chlorophenol under the catalytic action of peroxidase to form a red dyestuff (Trinder endpoint reaction). The color intensity of the red dyestuff formed is directly proportional to the triglyceride concentration and can be measured photometrically.
Triglycerides are fatty acid esters of glycerol that have three hydroxyl groups. Because they are insoluble in water, the triglycerides are transported with other more polar lipids. Elevated triglyceride measurements are associated with diabetes mellitus, pancreatitis, alcoholism, glycogen storage disease, hypothyroidism, nephrosis, pregnancy, use of oral contraceptives and gout. Triglyceride levels are decreased in hyperthyroidism, use of certain lipid-lowering drugs and malabsorption syndrome. Refer to the Laboratory Method Files section for detailed laboratory procedure manual(s) of the methods used.
One important factor to mention is that all blood draws for lipid analysis were collected on fasting participants tested in the morning. The sample size was accordingly reduced. This is now mentioned in text.
Reviewer’s follow-up to authors’ response: This is not the kind of information being requested. Readers as well as reviewers of this journal know what triglycerides are. Usually all covariates are well described in the methods section of a paper. Table 6 includes HDL, LDL, triglycerides, and blood pressure – all of which were never mentioned in the methods section in the first version of this manuscript. The new edition at lines 414-416 should be information provided in the methods section. These are not results. The new sentence at line 186-187 is not what was requested.
New comments
Line 13: The method used to collect the diet data is called the 24-hour dietary recall. Please use this terminology.
Line 18: According to table 1, tea consumption is highest for older adults 71+y.
Lines 110-116: NHANES participants were only stratified by age and gender? Again, all covariates used in the analyses should be described in the methods section. For example, education is not mentioned.
Line 116: A period is needed at the end of this sentence.
Line 151: The word calculate is spelled incorrectly.
Lines 155-157: The HEI doesn’t discourage or encourage, it is a scoring system that provides more points for certain components and less points for other components. Consider revising this to reflect that.
Lines 23-231: Suggest stating “Average amounts based on 2 days of dietary recalls/ participant (in grams/day) and …..”
Lines 237-239: Were both 24-hour dietary recalls used? Were the HEI scores averaged together for a 2 day mean?
Line 265: There is a period after 3.5.
Table 1: Are you certain that it is SD and not standard error listed with amounts grams? Usually when providing estimates of the population, standard errors are presented with the mean.
Write-up in results for figures 1, 2, and 3: Are the differences statistically significant?
Table 6: The estimates in table 6 should be clearly indicated – I am assuming they are mean and standard error?
Line 422-425: Where are the results of this correlation analysis presented? Or is this data not shown? All analyses that are stated or presented in the paper should be described in the methods section.
Line 435: NHANES is spelled incorrectly.
Line 436-439: These sentences are confusing. Not clear what is being stated here.
Line 503-508: Why is this relevant to this paper?
